# A platform for brain-wide imaging and reconstruction of individual neurons

Michael N Economo[1†], Nathan G Clack[1†], Luke D Lavis[1], Charles R Gerfen[1,2], Karel Svoboda[1], Eugene W Myers[1,3*‡], Jayaram Chandrashekar[1*‡]

[1]Janelia Research Campus, Howard Hughes Medical Institute, Ashburn, United States; [2]Laboratory of Systems Neuroscience, National Institute of Mental Health, Bethesda, United States; [3]Max Planck Institute of Molecular Cell Biology and Genetics, Dresden, Germany

**Abstract** The structure of axonal arbors controls how signals from individual neurons are routed within the mammalian brain. However, the arbors of very few long-range projection neurons have been reconstructed in their entirety, as axons with diameters as small as 100 nm arborize in target regions dispersed over many millimeters of tissue. We introduce a platform for high-resolution, three-dimensional fluorescence imaging of complete tissue volumes that enables the visualization and reconstruction of long-range axonal arbors. This platform relies on a high-speed two-photon microscope integrated with a tissue vibratome and a suite of computational tools for large-scale image data. We demonstrate the power of this approach by reconstructing the axonal arbors of multiple neurons in the motor cortex across a single mouse brain.

**\*For correspondence:** myers@ mpi-cbg.de (EWM); chandrashekarj@janelia.hhmi.org (JC)

[†]These authors contributed equally to this work
[‡]These authors also contributed equally to this work

**Competing interests:** The author declares that no competing interests exist.

## Introduction

A core goal of modern neuroscience is to map neural circuits across the entire brain, as long-range connectivity dictates how information flows between brain regions (*Bohland et al., 2009*). Brain-wide inter-areal connectivity has been probed using bulk injections of anterograde and retrograde tracers (*Gerfen and Sawchenko, 1984*; *Hunnicutt et al., 2014*; *Luppi et al., 1990*; *Markov et al., 2014*; *Oh et al., 2014*; *Veenman et al., 1992*; *Zingg et al., 2014*) and functional imaging methods with millimeter-scale spatial resolution (*van den Heuvel and Hulshoff Pol, 2010*). However, inter-mingled neurons within a brain region have heterogeneous projection patterns (*Nassi and Callaway, 2009*; *Shepherd, 2013*) and carry diverse messages in the behaving brain (*Li et al., 2015*; *Movshon and Newsome, 1996*; *Sato and Svoboda, 2010*; *Turner and DeLong, 2000*). Projection maps based on bulk tracer injections are thus comprised of multiple functionally and structurally distinct groups of neurons. Reconstructions of single neurons therefore provide vital information about how neural signals are organized and transmitted across the brain to target regions.

Single-neuron reconstructions are also critical for defining the fundamental cell types in the brain. The nervous system has long been viewed as a collection of discrete cell types defined by their developmental origins, gene expression, morphology, and function. Axonal structure has been particularly useful in classifying projection neurons (*Cowan and Wilson, 1994*; *Kawaguchi et al., 1990*) but this information is currently lacking for most cell types. Visualization of the brain-wide projection maps of the individual neurons comprising a projection pathway would enable the isolation of well-defined classes that target similar structures and may otherwise be impossible to distinguish.

Single-cell and sparse-labeling techniques (*Furuta et al., 2001*; *Horikawa and Armstrong, 1988*; *Pinault, 1996*; *Reiner et al., 2000*; *Rotolo et al., 2008*) have facilitated the reconstruction of individual axonal projections over long distances in the basal ganglia, hippocampus, neocortex, thalamus, olfactory cortex, and neuromodulatory systems (*Ghosh et al., 2011*; *Kawaguchi et al., 1990*;

**eLife digest** Nerve cells or neurons transmit electrical impulses to each other over long distances. These signals travel through highly branching nerve fibers called axons, which are about one hundred times thinner than a human hair, and can extend across the entire brain. Tracing the axon of a neuron from start to end can help to explain how individual neurons and brain areas communicate signals over long distances.

A mouse brain contains approximately 70 million neurons, and tracing the axons of many neurons within a brain is a challenging problem. Tackling this problem requires a method for imaging entire brains in high enough detail to unambiguously resolve and follow axons from individual neurons across the brain. Economo, Clack et al. now demonstrate such a method for three-dimensional imaging of tissue samples as large as the whole mouse brain.

This system is fully automated and works by first imaging a layer of tissue near the exposed surface of a sample, and then cutting off a slice of tissue that corresponds to the volume that has been imaged. These steps then repeat until the entire sample has been imaged; this takes about a week for a whole mouse brain and produces about 30 terabytes of images.

Economo, Clack et al.'s advance can uncover how neurons communicate over long distances with an unprecedented level of precision. The method can now be used to generate a comprehensive database of neurons and their long distance connections. Such a database would aid efforts to model the roles of neural circuits in the brain, and inform the design of experiments to study brain activity during particular behaviors.

*Kita and Kita, 2012*; *Kuramoto et al., 2009*; *Oberlaender et al., 2011*; *Ohno et al., 2012*; *Wittner et al., 2007*; *Wu et al., 2014*). However, the reliability and throughput of axonal reconstruction have remained limited by the necessity to restrict labeling to one or a few neurons in a single brain and to manually track individual segments between serial sections, which are often deformed and may be damaged by standard histological processing techniques.

Visualization of neurons in continuous whole-brain image volumes may overcome these limitations and allow reconstruction of long-range axonal projections in a reliable and efficient manner. This requires an approach capable of imaging every location in a three-dimensional space within a large tissue volume with high resolution so that fine axonal collaterals may be unambiguously tracked to their targets. Here, building on serial two-photon (STP) tomography (*Portera-Cailliau et al., 2005*; *Ragan et al., 2012*; *Tsai et al., 2009*), we describe a system for fast volumetric microscopy that can be used to image the entire mouse brain at high resolution. Combining this technique with high-intensity, sparse neuronal labeling, a novel technique for clearing tissue, and an informatics pipeline for processing and visualizing large imaging datasets, we present a platform for efficient reconstruction of axonal morphology. We demonstrate the utility of this system by reconstructing the extensive, brain-wide axonal arborizations of diverse projection neurons in the motor cortex within a single mouse brain.

## Results

### High-speed volumetric STP tomography

To image the axonal arbors of individual neurons, we constructed a platform for fast, automated, volumetric fluorescence imaging at sub-micron resolution. This system is based on a two-photon laser scanning microscope integrated with a vibrating microtome (*Figure 1a*), and was optimized for high-fidelity mosaic imaging in three dimensions.

To achieve high imaging speeds, we used a resonant scanning galvanometer, a piezoelectric motor-mounted objective for fast Z-scanning, and high excitation power. This approach allowed us to image up to 16 million voxels/s (>50 mm$^3$/day) with signal-to-noise ratios (SNRs) sufficient for axonal reconstruction. This fast-scanning microscope represents a 16–48$\times$ increase in speed over conventional laser-scanning microscopy systems (*Pologruto et al., 2003*). The system is comprised primarily of commercially-available hardware and custom control software (freely available at github.

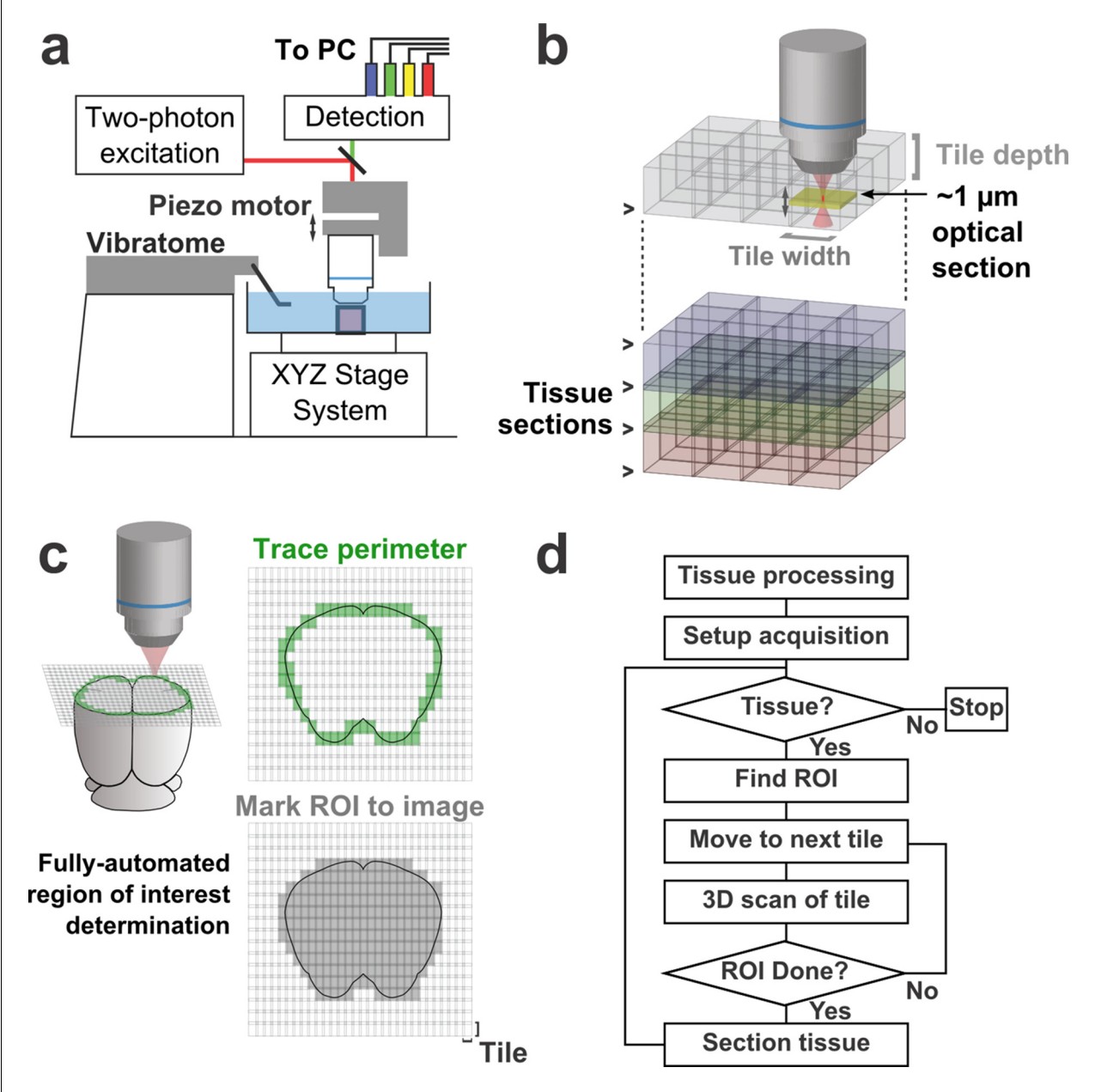

**Figure 1.** Schematic of imaging system. (a) Schematic of apparatus for automated volumetric two-photon tomography. (b) To image large volumes of tissue, a collection of three-dimensional image stacks (tiles) covering the full volume of the tissue sample was acquired serially. Tiles overlapped in all three dimensions to aid in image registration and ensure coverage of the full volume. (c) The active imaging region is determined for each section by first tracing the perimeter of the tissue block and then filling in any tiles internal to the traced region. (d) Flow chart illustrating the tasks performed during data acquisition.

The following figure supplement is available for figure 1:

**Figure supplement 1.** Point spread function measurement.

com/TeravoxelTwoPhotonTomography) and operates as follows: (1) the entire volume of space addressable by the stage system is partitioned into overlapping 3D tiles, each corresponding to a single image stack (*Figure 1b*); (2) tiles intersecting the exposed surface of the sample are marked for imaging (*Figure 1c*); (3) an image stack corresponding to each marked tile is acquired; (4) the integrated vibratome removes a portion of the imaged volume; (5) the precise surface plane of the

remaining tissue is determined; and (6) this sequence of events is automatically repeated until the entire tissue sample has been processed (*Figure 1d*).

## Sample clearing and embedding

In the past, STP tomography has been limited to acquiring two-dimensional (2D) snapshots of tissue volumes separated by >50 µm in the axial direction. Following small structures through highly scattering brain tissue in three dimensions requires a strategy for clearing fixed tissue that is compatible with long-term imaging. Specifically, we sought an approach that preserved the native fluorescence of fluorescent proteins, introduced little background autofluorescence, effectively cleared gray and white matter, was compatible with serial sectioning, and that could be used as a stable, long-term immersion medium during extended imaging sessions. We evaluated the compatibility of a number of recently described techniques (*Chung and Deisseroth, 2013*; *Ertürk et al., 2012*; *Hama et al., 2011*; *Ke et al., 2013*; *Renier et al., 2014*; *Susaki et al., 2014*). Strategies employing organic solvents (i.e. 3Disco, iDisco) quenched fluorescent proteins produced hard, brittle tissue that could not be sectioned. Clearing agents requiring high-index saturated solutions (i.e. SeeDB, Clarity, CUBIC) were not robust to evaporation and solutes precipitated during extended imaging experiments. Sc*ale* expanded tissue, attenuated fluorescence, and resulted in unstable tissue geometry. Due to these incompatibilities, we sought an alternative method meeting these requirements.

We devised a clearing medium comprised of a ternary mixture of dimethyl sulfoxide (DMSO), D-sorbitol, and aqueous buffer (see Methods). DMSO was chosen due to its high refractive index (1.479) and extensive use in mounting medium formulations for fluorescence microscopy. The sugar alcohol D-sorbitol was found to be highly soluble in DMSO and, unlike D-fructose (*Ke et al., 2013*), may be obtained at high purity without autofluorescent contaminants and does not cause non-enzymatic browning of biological tissue. Aqueous buffer was added to preserve green fluorescent protein (eGFP) fluorescence (*Figure 2b,c*). The high refractive index of the resulting solution (1.468) reduced light scattering in tissue by minimizing index mismatches with lipid- and protein-rich brain structures (*Figure 2a*). The excellent tissue penetrance of these reagents further contributed to a high degree of clearing of white and gray matter in whole brains. We note that the use of subsaturating D-sorbitol solutions was important for preventing precipitation or gellation of the mixture following the evaporation and mechanical agitation that accompanies long (>1 week) imaging sessions. Another key advantage of this medium over other clearing agents is its low viscosity; serial sections with consistent thickness were difficult to obtain in viscous media. This approach allowed small structures, including fine axons, to be imaged with minimal signal attenuation at imaging depths >200 µm (*Figure 2d–f*). Samples could be reliably sectioned and imaged over the course of many days and stored for many months without adverse effects on fluorescent labels or gross changes in brain volume (*Figure 2a*). Hence, this medium met the requirements for high-resolution volumetric imaging and serial sectioning.

To reduce distortion of the tissue during sectioning, samples were embedded in gelatin and fixed a second time. This procedure ensured adhesion of the tissue to the surrounding gelatin. As our clearing solution quickly diffused through the cross-linked gelatin matrix, tissue samples—including whole mouse brains—could be effectively cleared following gelatin embedding. Treatment with CUBIC-1 (*Susaki et al., 2014*) for lipid removal prior to embedding was found to further improve clearing (data not shown). The complete protocol for preparation of cleared tissue samples for long-term imaging is detailed in *Materials and methods* and *Table 1*.

## High-resolution whole-brain imaging

The diameters of fine axonal processes are often less than 100 nm (*De Paola et al., 2006*; *Shepherd and Harris, 1998*). A complete volumetric representation of axonal arborizations thus requires diffraction-limited, high-numerical aperture (NA) imaging and near-Nyquist sampling (voxel size ∼ 0.3 × 0.3 × 1.0 µm). At this resolution, the full volume of a mouse brain contains more than 5 trillion voxels, corresponding to more than 10 TB of image data per fluorescence channel. In addition, physical sectioning introduces tissue deformation at the micron scale (*Figure 3*). For these reasons, it was necessary to construct a computational pipeline for registering, visualizing and storing datasets of this magnitude.

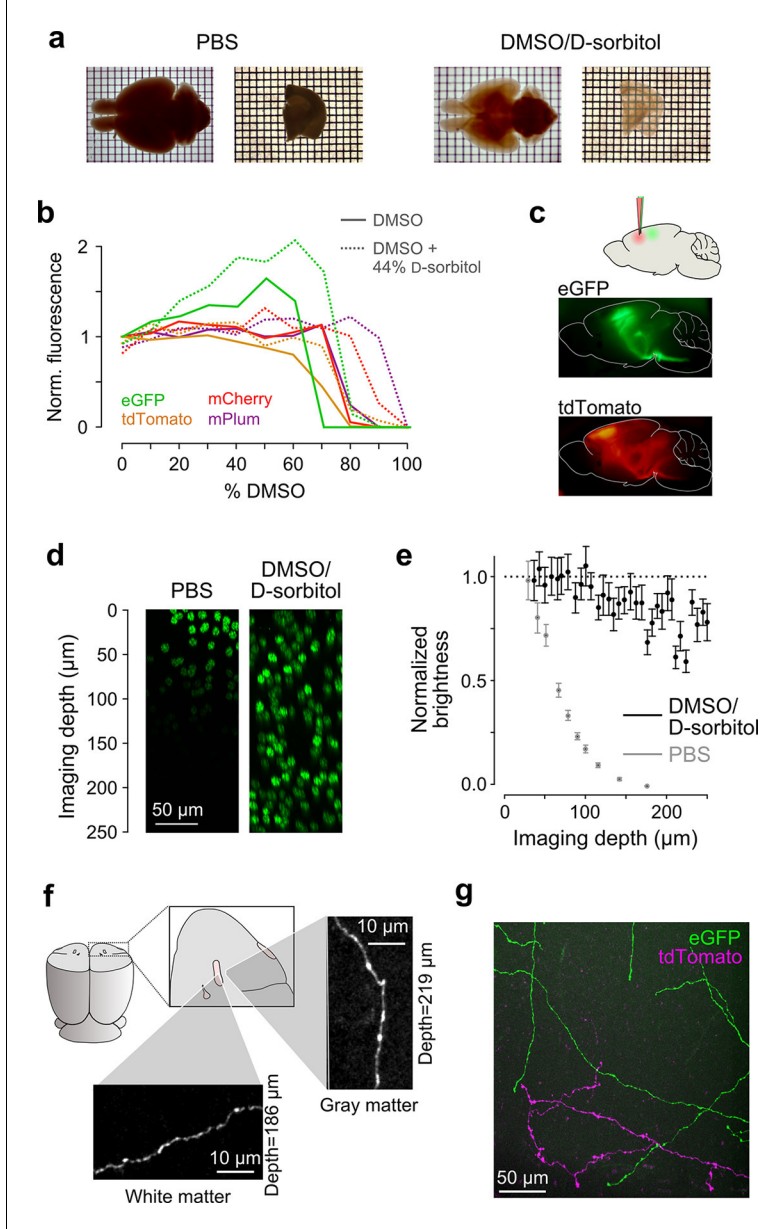

**Figure 2.** Sample preparation and clearing. (a) Whole brains (*left*) and 1 mm-thick tissue sections (*right*) cleared using dimethyl sulfoxide (DMSO) and D-sorbitol. (b) Fluorescence of purified eGFP as a function of DMSO concentration (v/v) in 10 mM HEPES, pH 7.3 (open circles) and 10 mM HEPES, pH 7.3 containing 44% (w/v) D-sorbitol (filled circles). (c) eGFP and tdTomato fluorescence in a cleared hemi-brain demonstrates preservation of native fluorescence. (d) Maximum intensity projections of side views of image tiles of histone 2B-eGFP labeled nuclei from a DMSO/D-sorbitol cleared section (*right*) and from a matched section in phosphate buffered saline (PBS*; left*) from the contralateral hemisphere of the same brain. (e) Quantification of intensities of histone 2B-eGFP labeled nuclei as a function of depth (PBS: n=204 detected nuclei in two tiles; DMSO/D-sorbitol: n=815 detected nuclei in three tiles). (f) Representative images of a single axonal collateral as it passes through the anterior commissure (white) and surrounding olfactory cortex (gray). Images were acquired at a depth >180 μm in each case and scaled in the same manner. (g) Axons labeled with multiple fluorophores could be simultaneously imaged with high signal to noise within cleared tissue. Note that imaging fine axons through the axially oriented anterior commissure in this example represent a stringent test of clearing due to the high degree of optical scattering observed in white matter tracts. DMSO, dimethyl sulfoxide; eGFP, enhanced green fluorescent protein.

**Table 1.** Clearing solutions.

| Solution (#) | Dimethyl sulfoxide (g) | PB (g) | D-sorbitol (g) | Refractive index |
|---|---|---|---|---|
| 1 | 26.83 | 73.17 | 0.00 | 1.373 |
| 2 | 52.38 | 47.62 | 0.00 | 1.412 |
| 3 | 44.52 | 40.48 | 15.00 | 1.425 |
| 4 | 36.67 | 33.33 | 30.00 | 1.440 |
| 5 | 34.25 | 20.75 | 45.00 | 1.468 |
| 6 | 25.56 | 11.44 | 63.00 | 1.489 |

To enable registration of tiles with submicron resolution, we ensured that they partially overlapped in all dimensions. Following image acquisition, a set of common features was identified in the overlap region of each pair of tiles bordering each other in the axial direction (i.e. tiles at the same x, y location in successive sections). Autofluorescent puncta (lipofuscin; *Figure 3—figure supplement 1*) were present throughout the mouse brain and could be reliably identified in overlapping tiles. Using these point correspondences, the tiles were registered (*Materials and methods*; *Figure 3*) and resampled into a set of non-overlapping image volumes precisely tiling the full brain. The resampled volume was hierarchically downsampled to produce additional representations of the volume at progressively lower resolution. This multiresolution volume enabled efficient navigation at different spatial scales and neuronal structures, which appeared continuous across tile boundaries (*Video 1*), could be traced for over long distances.

To test the imaging and informatics pipeline, we imaged whole mouse brains wherein a sparse subset of neurons in motor cortex was brightly labeled (*Figure 4*). Sparse labeling was achieved by expressing Cre recombinase with a highly diluted adeno-associated virus while simultaneously delivering a Cre-dependent high-titer reporter virus coding for a fluorescent reporter (e.g. eGFP) (*Xu et al., 2012*). Using this labeling strategy, 10–50 motor cortical neurons could be routinely labeled with high intensity in each brain. Brightly-labeled neurons were likely infected with a single copy of the viral plasmid encoding Cre but with many copies of the reporter gene, thereby driving high expression in few cells. High-intensity labeling (e.g. with viral vectors encoding bright fluorescent proteins like eGFP or tdTomato) was necessary for high-contrast imaging of thin fibers. Using this labeling strategy, fine, long-range axonal collaterals—such as those innervating the contralateral motor cortex—were easily detected and resolved. These fibers were clearly discernable even in a maximum intensity projection through a large volume of tissue ($2.3 \times 1.3 \times 1.0$ mm) corresponding to >200 individual 3D tiles (*Figure 4b*). The high excitation power needed to image axons with high SNR was not found to have a detrimental effect on the resolution of our imaging system due to fluorophore saturation, as has been suggested (*Figure 4—figure supplement 1*) (*Zipfel et al., 2003*).

## Reconstruction of long-distance axonal morphology

Whole-brain reconstruction of axon arborizations requires that fine-caliber segments far from their associated somata remain brightly labeled. We examined axonal labeling at distant locations for five of the cells depicted in *Figure 4b* (somatic location and dendritic morphology illustrated in *Figure 5a*). Axonal segments associated with brightly labeled somata remained well-labeled through white matter tracts, at branch points, and at their termini in distant structures, indicating high labeling intensity throughout each arbor (*Figure 5b*). We quantified the SNR across 52 axonal cross sections selected to represent a range of axonal morphologies. In this sample, the z-scored median peak intensity of individual axons was consistently high compared to background ($15.0 \pm 2.9$ median $\pm$ standard error of the mean; 5–95% range: 7.0–70.7; *Figure 5—figure supplement 1*). Further, we found that datasets could be stored using lossy H.264 compression while maintaining high image quality. At a compression ratio of 25:1, the SNR of small axons was reduced by only 4.8% (compression ratio determined in n = 996 tiles; *Figure 5—figure supplement 1*). Software tools for data registration, resampling, and compression are available online at github.com/TeravoxelTwoPhotonTomography.

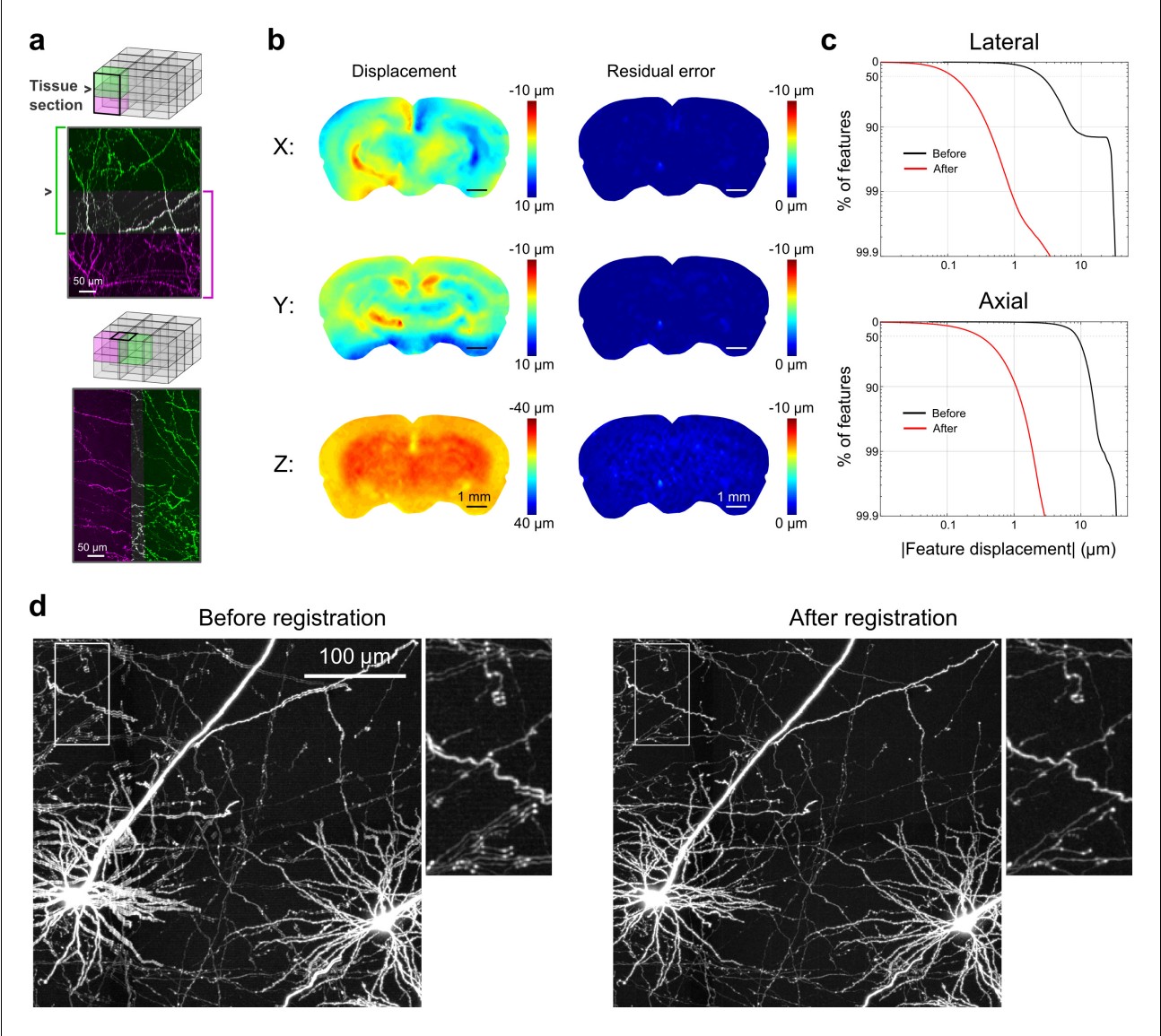

**Figure 3.** Registration of image tiles. (**a**) Example registration of pairs of image tiles in the axial (left) and lateral (right) directions. (**b**) Initial displacement of automatically-identified features as a result of sectioning (*left*) across the full extent of the exposed tissue surface. Displacements along each Cartesian direction are displayed in separate heat maps for a representative section. *Right:* residual displacement of the same feature set after linear interpolation of displacements across each tile. (**c**) Distribution of the residual displacements of all features identified in a whole-brain dataset in the lateral (*top*) and axial (*bottom*) directions before (black) and after (red) image registration. (**d**) Maximum-intensity projection through a volume containing labeled neurons before (*left*) and after (*right*) the registration procedure.

The following figure supplement is available for figure 3:

**Figure supplement 1.** Lipofuscin imaging.

Another requirement for axonal reconstruction is that single fibers must be tracked unambiguously across long distances. Even with sparse labeling, axonal segments of similar caliber but originating from different somata cross paths with separations smaller than the diffraction limit of light, introducing uncertainty in the assignment of axonal identity following crossover points. In the past, the inability to unambiguously follow single processes has precluded extensive reconstruction of fine axonal collaterals in datasets containing multiple labeled neurons in the same brain region (*Gong et al., 2013*; *Zheng et al., 2013*). Since the error rate in manual and automatic reconstruction of axonal processes from light microscopy datasets depends strongly upon the density of neurites

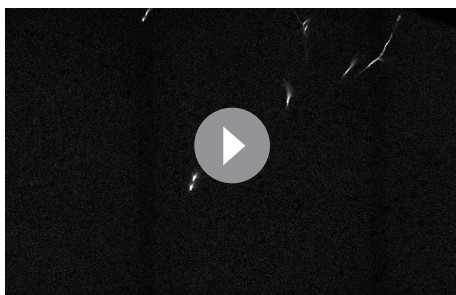

**Video 1.** Movie illustrating 18 serially acquired image tiles that have been registered and resampled into a continuous image volume. This volume (3 × 1 × 6 tiles) spans five adjacent tissue sections. Fine axons are resolvable and all fibers appear continuous. Images were spectrally unmixed to remove autofluorescence of lipofuscin.

within a local volume (*Gala et al., 2014*), we assessed the feasibility of complete axonal reconstruction in this dataset. We attempted to reconstruct the full axonal morphology of the same cohort of neurons depicted in *Figure 5*. The high density of processes at the site of viral injection - the region containing the highest density of labeled somata - prevented unambiguous reconstruction of many processes in this area due to axonal crossovers (*Figure 5c*). However, we found that the long-distance axonal projections of these neurons and their fine collaterals in target regions could be accurately reconstructed to a high degree of completeness (*Figure 6a*; *Video 2*). This level of completeness was facilitated by the labeling sparsity, high resolution imaging, and lack of discontinuities in the dataset. Furthermore, 3–12 mm of axon could be reconstructed per hour, depending on local density—a substantial increase in throughput over classical serial-section reconstruction methods, even when samples contained only a single labeled neuron (*Wittner et al., 2007*).

This group of reconstructed neurons included two intratelencephalic (IT) projection neurons in layer II/III, two IT neurons in layer V, and one corticothalamic projection neuron located in layer VI (*Figure 6a*; *Videos 3–5*). These neurons possessed extensive axon collaterals with each cell targeting a surprisingly diverse set of brain areas (*Table 2*, *Figure 6a*). Together, the total axon length of these five neurons exceeded 300 mm (range: 23.6–121.2 mm) and innervated 28 distinct brain regions (*Table 2*). The arborization of one individual layer V neuron was found to cover nearly the entire extent of the brain in the coronal plane (*Figure 6b*) and approximately half of the brain along the rostrocaudal axis, underscoring the importance of whole-brain imaging for determining axonal projection patterns. Interestingly, this neuron included an extensive projection to the taenia tecta (*Figure 6b*), a region that exhibits only very sparse axonal labeling in bulk anterograde tracing experiments from motor cortex (C. Gerfen, unpublished data; Allen Connectivity Atlas: connectivity. brain-map.org/projection/experiment/141603190), further highlighting the diversity in projection patterns of similar neurons within a single brain region. Following our approach, we were able to reconstruct axon morphologies to a high degree of completeness. As a result, the ensemble of downstream targets could be precisely identified for single neurons, and, furthermore, the extent and location of collaterals in all target regions.

Importantly, reconstructing multiple neurons within a single brain permits fine-scale comparisons between individual neurons. For example, in this limited dataset, all four reconstructed IT neurons projected to the contralateral motor cortex. The contralateral cortical projections of the two layer V neurons spanned a similar distance along the anterior–posterior (A–P) axis (red: 2152 um; black: 2096 um). However, their terminal fields were offset along this axis by an amount closely mirroring the A-P offset between their corresponding somata (*Figure 7a,b*; distance between projection centroids = 490 μm, A-P distance between somata = 513 um). This relationship suggests that these projections might target a continuum of cortical locations rather than a single discrete structure. In contrast, the contralateral cortical projection of both layer II neurons targeted a similar location along the A-P axis more posterior than both somata (*Figure 7c,d*; distance between projection centroids = 32 μm; A-P distance between somata = 271 μm). Although a detailed analysis of such topology in a larger population is beyond the scope of this study, this example illustrates how organizational principles in neural connectivity—even at the micron scale—can be investigated using our approach.

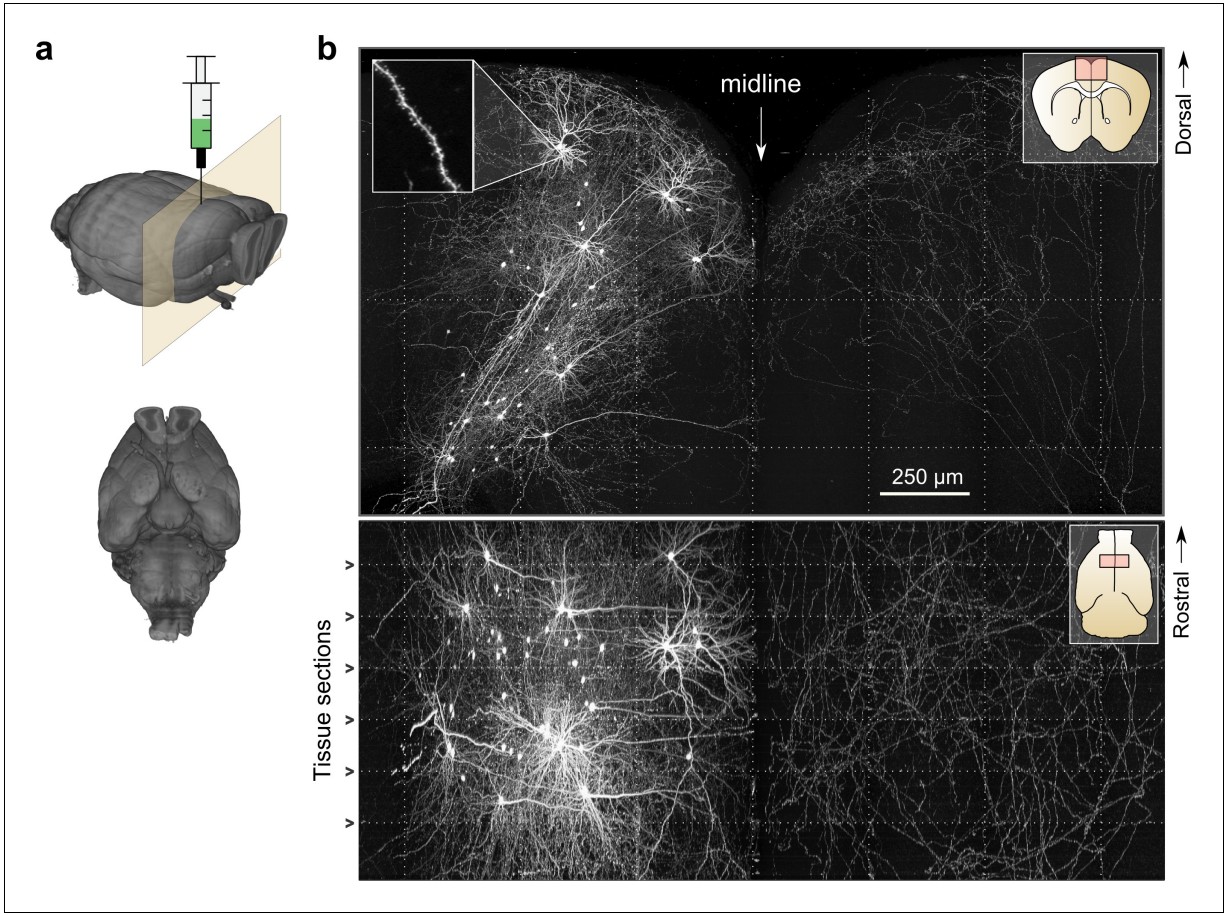

**Figure 4.** Whole-brain imaging. (a) Three-dimensional rendering of complete mouse brain dataset as viewed from an anterolateral (*left*) and ventral (*right*) perspective. (b) Maximum intensity projection through a large tissue volume containing labeled somata and neurites in the X–Y (*top*) and X–Z (*bottom*) planes (2.3 × 1.3 × 1.0 mm). Axon collaterals are clearly discernible in the contralateral hemisphere despite the large volume of tissue depicted. Dotted lines represent borders between adjacent tiles. Insets (*top right*) represent the location and orientation of each image in relation to a coronal (*top*) and horizontal (*bottom*) section. Inset at top left illustrates detail in full-resolution images. Following registration of the full collection of tiles comprising a full dataset, individual neurites appear continuous between tiles separated in the axial direction (*bottom*; separated by a physical tissue section) while maintaining continuity in laterally adjacent tiles (*top*).

The following figure supplement is available for figure 4:

**Figure supplement 1.** High speed, high-power imaging.

## Discussion

Visualization of the complete axonal morphologies of individual neurons is critical for understanding how diverse neural signals are organized and communicated to distant targets throughout the brain. We have developed an imaging system for whole-brain, high-resolution fluorescence imaging and demonstrate that this approach can be used to trace individual axonal fibers and fine axon collaterals across the brain to their termini (*Video 2*) and thereby map the long-distance projections of single neurons (*Figure 6*).

Reconstruction of axonal processes with light microscopy requires 3D, diffraction-limited imaging and a high-NA optical system in order to effectively identify and disambiguate axons originating from different cells. At the same time, light scattering within tissue must be minimized in order to image full 3D volumes. To achieve high resolution with minimal scattering, we extended block-face STP tomography (*Portera-Cailliau et al., 2005*; *Ragan et al., 2012*; *Tsai et al., 2003*) to high-speed 3D imaging. This required the development of a compatible procedure for tissue clearing, construction of an automated imaging system that can reliably image large 3D volumes, and design and

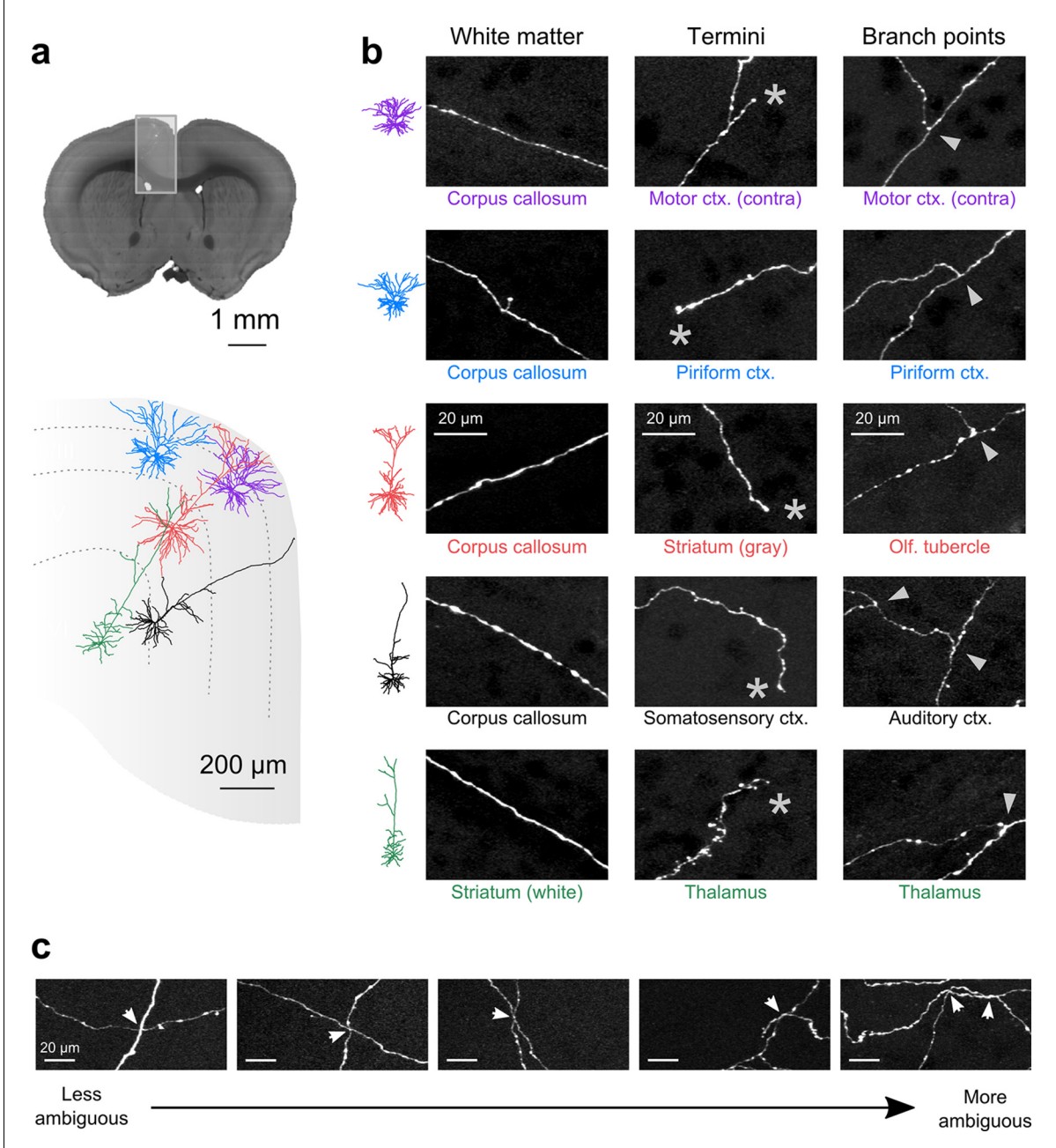

**Figure 5.** Axon collaterals are labeled with high signal-to-noise across their entire length. (**a**) *Top:* Virtual coronal section through whole-brain dataset. Boxed area denotes region containing labeled somata and is expanded in schematic below. *Bottom:* Laminar distribution and dendritic morphology of five labeled neurons spanning layers II–VI. (**b**) Representative images of axonal segments from each of the five neurons in white matter (*left*), at branch points (*middle;* denoted by asterisks) and at termini (*right;* denoted by arrowheads). The location of each axonal segment is listed below the corresponding images, and in all cases, segments were in a different brain structure than their associated somata. All images are maximum intensity projections through a depth of 20 μm. (**c**) Regions with high labeling density contain axon crossovers that introduce ambiguity into axonal reconstruction. Examples of crossover points ordered subjectively by approximate degree of difficulty in assignment of segment identity. The two-dimensional examples shown here are for illustration purposes; ambiguity arises only when segments are closely apposed in all the three dimensions.

The following figure supplement is available for figure 5:

**Figure supplement 1.** Signal-to-noise of axonal imaging.

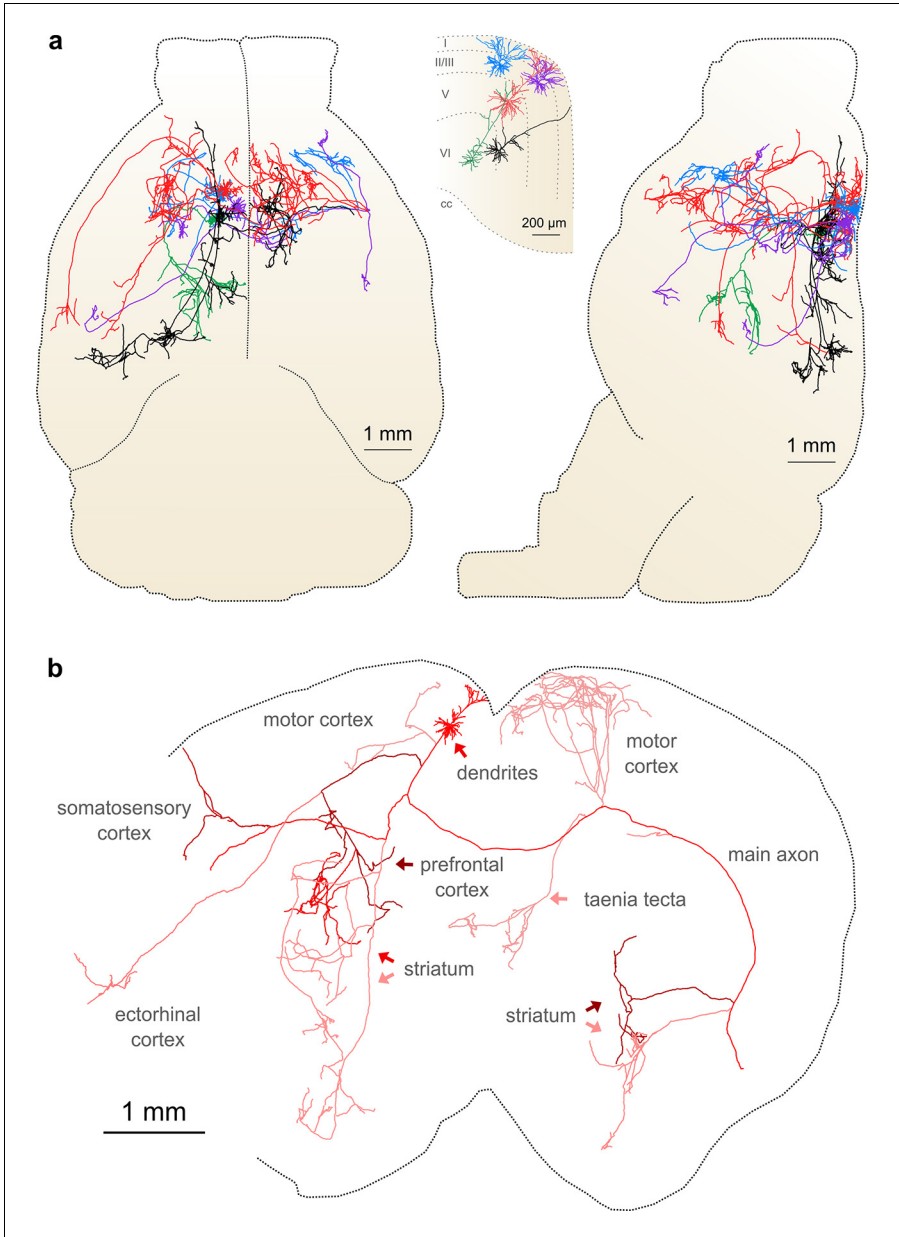

**Figure 6.** Complete reconstruction of axonal morphology. (**a**) Complete reconstruction of the same five projection neurons depicted in *Figure 4* (inset). Reconstructions are overlaid on a horizontal (*left*) and sagittal (*right*) outline of the imaged mouse brain. Subset includes pyramidal neurons in layer II (blue, purple), layer V (red, black) and layer VI (green). (**b**) Illustration of axonal and dendritic reconstruction of the layer 5 pyramidal cell (colored red in *a*) shown in the coronal plane. Profile of coronal section at the rostrocaudal position of the cell body is depicted by the black dashed line. Segments were colored to highlight axonal arbors originating from common branch points.

construction of novel computational tools for the registration and visualization of large (up to 100 TB) datasets. In previous studies, STP tomography has been used to image bulk axonal projections across the mouse brain (*Oh et al., 2014*). However, this approach has been limited to the acquisition of sequences of 2D images spaced at large intervals; high-SNR imaging of complete, 3D volumes - a necessity for tracing continuous neurites - has not been previously achieved using this approach.

A number of alternative modalities are also potentially well-suited to high-resolution whole-brain imaging, although none have been successfully applied to extensive, brain-wide reconstruction of axon collaterals. Selective plane illumination microscopy (SPIM) (*Dodt et al., 2007*; *Huisken et al., 2004*) requires reduced acquisition times, but lacks the necessary spatial resolution (< 0.5 µm) for

resolving close appositions between thin axonal collaterals within large tissue volumes. Knife-edge scanning microscopy ( *Li et al., 2010*; *Mayerich et al., 2008*) represents a potential alternative, but requires resin embedding of tissue samples that can quench or attenuate the fluorescence of native fluorophores and produces tears in large tissue volumes along the edges of 'ribbon' sections. While some axonal segments have been traced using this approach, it has been unable to identify the vast majority of fine axon collaterals innervating many target structures (*Gong et al., 2013*; *Zheng et al., 2013*). Large-scale electron microscopy (*Kleinfeld et al., 2011*) offers the potential for dense reconstruction of neurites and synapses, but is not yet feasible for volumes larger than 1% of the mouse brain—precluding the reconstruction of long-range axonal projections. The resolution and imaging speeds reported for alternative light- and electron-microscopy technologies are summarized in *Table 3*.

Sparse neuronal labeling is a fundamental requirement for the reconstruction of axon collaterals using light microscopy. Ambiguity is introduced when axons originating from different somata are juxtaposed with separations less than the resolving limit of the optical system (*Figure 5c*). This challenge can be mitigated by imaging with the highest available resolution and imposing a high degree of labeling sparsity—but cannot be eliminated entirely. In the example dataset presented here, excessive labeling density was largely limited to the site of virus injection and therefore only precluded the extensive reconstruction of local fibers. Biological questions focused on local collaterals will require alternative labeling strategies that achieve greater separation between labeled somata to mitigate this limitation. Sparse stochastic genetic methods (*Rotolo et al., 2008*) and retrograde labeling (*Mazarakis et al., 2001*; *Wickersham et al., 2007*) provide straightforward approaches for labeling well-separated somata.

Reconstructing multiple neurons within a single brain provides a particularly rich description of axonal organization. The spatial overlap - or, alternatively, segregation - of collaterals of different neurons within a single target region can be directly assessed at the micron scale. Comparison of projections of multiple neurons within the same population provides a means to better understand if topological organization in source areas is preserved in downstream structures. In addition, the fine-scale spatial relationships between axon collaterals from different sources may indicate how disparate streams of information remain isolated or are integrated in target structures. Such comparisons are challenging or impossible in datasets in which single neurons are reconstructed in different samples.

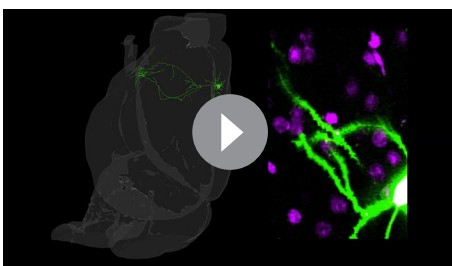

**Video 2.** Single axon traced to its terminus. Depicted path represents the longest continuous axonal segment starting at the cell soma. The terminus is located in the anterior piriform cortex. White dot corresponds to the same location in both panels and corresponds approximately to the path of manually traced segment. The SNR of well-labeled axons remains high along their entire length and branch points are clearly visible. Nuclei (magenta) were counter-labeled using NuclearID-Red. Two axons, originating from distinct neurons, cross in the corpus callosum. Axon identity is straightforward to assign in this example on the basis of trajectory and labeling intensity when viewed in three dimensions. This neuron appears blue in all figures.

Currently whole brain imaging still requires significant time (8–10 days) and resources (many TB of storage space). Labeling techniques that take advantage of multiple, spectrally separable fluorescent proteins (*Cai et al., 2013*; *Shaner et al., 2005*) and the targeting of several populations with non-overlapping projections in a single brain should permit labeling of hundreds or even thousands of neurons in each brain while maintaining sufficient sparsity for single neuron reconstruction (*Gala et al., 2014*). In addition, the high degree of automation employed in the acquisition procedure provides the opportunity for further scalability by taking advantage of multiple imaging systems operating in parallel. Although the time required for manual tracing remains substantial, this constraint is likely to be greatly reduced as a result of recent developments in automation of neuronal reconstruction (*Gala et al., 2014*; *Jain et al., 2010*; *Peng et al., 2014*) and the high SNR of axon imaging achieved with our approach.

Recent advances in imaging techniques, genetic targeting methods, and 'big data' tools have facilitated multiple, comprehensive large-

**Table 2.** Axonal reconstructions. All lengths in mm.

| Soma location | Layer II | Layer II | Layer V | Layer V | Layer VI |
|---|---|---|---|---|---|
| Dendritic branches | 66 | 57 | 37 | 25 | 23 |
| Dendritic length | 8.03 | 7.67 | 9.17 | 5.06 | 4.20 |
| Axonal branches | 79 | 67 | 178 | 136 | 27 |
| Axonal length | 47.34 | 41.04 | 121.20 | 75.44 | 23.57 |
| Axonal targets (ipsilateral) | Motor cortex<br>Dorsal striatum<br>Nucleus accumbens | Motor cortex<br>Orbital cortex<br>Dorsal/medial striatum<br>Caudal/lateral striatum | Motor cortex<br>Somatosensory cortex<br>Auditory cortex<br>Orbital cortex<br>Agranular insular cortex<br>Ectorhinal cortex<br>Piriform cortex<br>Dorsal striatum<br>Nucleus accumbens<br>Olfactory tubercle<br>Taenia tecta | Motor cortex<br>Somatosensory cortex<br>Auditory cortex<br>Anterior cingulate cortex<br>Posterior parietal cortex | Motor cortex<br>Thalamic nuclei:<br>Ventral anterior lateral<br>Posterior<br>Submedial<br>Reticular |
| Axonal targets (contralateral) | Motor cortex<br>Insular cortex<br>Piriform cortex | Motor cortex<br>Anterior cingulate cortex<br>Agranular insular cortex<br>Claustrum<br>Basolateral amygdala | Motor cortex<br>Piriform cortex<br>Dorsal striatum<br>Nucleus accumbens<br>Olfactory tubercle<br>Taenia tecta | Motor cortex<br>Anterior cingulate cortex<br>Dorsal striatum | |

scale efforts attempting to map neural connectivity. In model organisms, these projects have largely focused on mapping local connectivity using electron microscopy (*Bock et al., 2011*; *Helmstaedter et al., 2013*) and the bulk projections of neural populations using light microscopy (*Mitra, 2014*; *Oh et al., 2014*; *Zingg et al., 2014*). In humans, atlases of gross physical and functional connectivity have been constructed from diffusion-tensor imaging and resting-state functional MRI datasets (*Mori et al., 2009*; *van den Heuvel and Hulshoff Pol, 2010*). However, techniques for atlasing the brain-wide axonal collateralization of single mammalian neurons remain unavailable, even though single-cell reconstruction has long been the gold standard for establishing the structure of long-range projections. The approach described here for the efficient reconstruction of single-cell axon collateral maps represents a crucial step towards filling this gap in the available tools for mapping neural connectivity, and will contribute to the ongoing renaissance in large-scale neuroanatomy.

## Materials and methods

### Microscope design

We constructed a microscope for fast volumetric STP tomography. The mouse brain, approximately 0.5 cm$^3$ in volume, requires $5 \times 10^{12}$ voxels for a full representation at a voxel size of 0.1 μm$^3$. Acquiring this volume of data with conventional galvanometric scanning at a rate of 1 megavoxel/s would require about 4 months. We have developed a custom resonant scanning system capable of sustaining 16 megavoxels/s for up to 4 channels (128 MB/s data rate) to reduce imaging time to approximately 1 week per sample.

The scan system is composed of a resonant scanning mirror (CRS-8kHz, Cambridge Technology, Bedford, MA) and a 5-mm linear scanning mirror (6215H, Cambridge Technology) conjugated with a scan telescope comprised of two custom scan objectives (Special Optics, Wharton, NJ) to ensure flat scanning across the available scan field. To avoid thermally damaging the sample at the edge of the resonant mirror scan field, an adjustable slit was placed at a conjugate image plane in the excitation path to block laser light in the appropriate area. A Pockels Cell (302RM, Conoptics, Danbury, CT) was used to modulate laser power and provide fast shuttering between frames.

In this study, all imaging was performed using a 40×/1.3 NA oil-immersion objective (Objective Plan-Apochromat 40×/1.3 DIC, Carl Zeiss, Oberkochen, Germany). The emitted light was passed

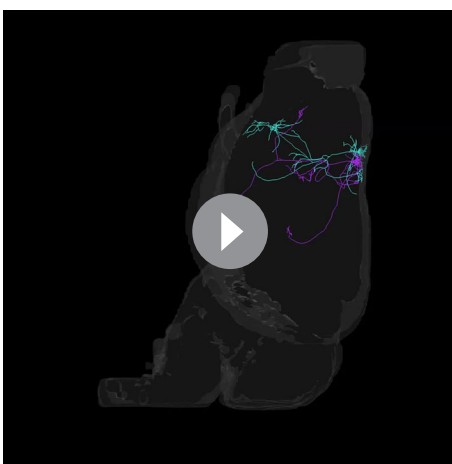

**Video 3.** Three-dimensional rendering of reconstructed Layer II motor cortical neurons. Displayed brain outline corresponds to the contours of the imaged brain. Color code is the same as in all figures.

through a primary dichroic (FF735-Di01-25×36, Semrock, Rochester, NY) onto photomultiplier tubes (H7422P-40, Hamamatsu Photonics, Hamamatsu, Japan) using a custom detection head (part of the Janelia MIMMS microscope design) (*Flickinger et al., 2010*). Two spectral channels were collected. eGFP emission was further filtered through a secondary dichroic (FF01-750/SP-25, Semrock) and an emission filter (FF03-525/50-30-D, Semrock). A second channel (FF02-617/73-30-D, Semrock) was collected for identification of red-labeled nuclei and/or spectral disambiguation of eGFP and broad-spectrum autofluorescence.

High-fidelity synchronization of the digitizer and scan system (Pockels cell, galvanometric scanner, and objective positioner) with the resonant mirror scan period was accomplished by reflecting a probe laser (635 nm, S1FC635PM, ThorLabs, Newton, NJ) from the resonant mirror onto a photodiode (PDA100A, Thorlabs). The photodiode was positioned 1 m away at one edge of the arc swept out by the resonant mirror. The voltage pulse generated by the incident probe laser was converted into a square wave using a custom circuit. This circuit was designed so that the rise of the square wave corresponds with the peak of the pulse observed from the photodiode, and permits adjustment of the clock phase in 5 ns increments.

Collected photons were converted to an analog photocurrent, which was then amplified and converted to an analog voltage using a custom transimpedance amplifier (impulse response FWHM=25 ns), and digitized using a 12-bit high-speed digitizer at 125 Msamples/second (ATS-9350, Alazar-Tech, Pointe-Claire, Canada). A voltage bias was applied at the amplifier to take advantage of the digitizer's full range. The digitizer generated records 16k samples wide at a rate of 8 kHz (256 MB/s/channel). These lines were distorted due to the spatially non-linear motion of the resonant mirror. Because of this, they were resampled using a cosine lookup table in a manner that is photoefficient (discards no samples) and avoids aliasing. Resampling was implemented on a GPU (GTX 580, NVidia, Santa Clara, CA) in order to meet data throughput requirements.

Scanning a large volume of tissue was achieved by serially scanning a collection of smaller 3D stacks (tiles). Each of these tiles overlapped adjacent neighbors along each dimension, permitting post hoc registration. To address each tile, the sample was translated in three dimensions using a high-precision mechanical stage system (XY: M-511.DD, Z: M-501-DG, Controller: C843; Physike Instrumente, Karlsruhe, Germany). Fast scanning in the Z-direction was achieved using a piezo-electric motor to axially translate the objective (P-725K.103 and E-665.CR, Physike Instrumente). The maximum depth of each tile - typically set to 250 μm—was limited by the working distance of the objective and the range of the objective piezo.

Integrated tissue sectioning was implemented on a custom assembly built around the vibrating servo from a Leica 1200S (Leica Microsystems, Wetzlar, Germany). Vibration was driven using an 86 Hz sine wave generated from a function generator and custom amplifier with vibration amplitude of approximately 1 mm. Tissue samples were positioned and fed across a stationary vibrating blade using the stage system at a velocity of 0.1 mm/sec. In this way, tissue sections >25 μm could be reliably removed from the tissue block once they were imaged. Computer aided design drawings and bill of materials are available at github.com/TeravoxelTwoPhotonTomography.

## Microscope operation

Fully automated microscope operation was controlled by custom software (available at github.com/TeravoxelTwoPhotonTomography). This software incorporates several features that ensure reliable operation spanning days to weeks. First, to guarantee an overlapping volume is reimaged after sectioning, the acquisition software applies a surface-finding procedure and adjusts the focal plane to

ensure that tiles begin at the tissue surface. In this procedure, the surface is identified by searching for the top of the tissue by imaging autofluorescence in a subset of tiles with a coarse Z spacing around the expected surface position. Second, algorithms monitor the data quality of the incoming video stream to detect, for example, a photodetector fault. Third, to ensure that high data rates can be sustained over long acquisition times with high fidelity, data are locally cached on a solid-state hard drive (840 pro; Samsung) and asynchronously transferred to a highly-redundant network file system. Fourth, in the event of a fault, operation is safely halted and may resume from the point at which the fault occurred.

Volume imaging is completed in a fully automatic process. After the top plane of the sample is positioned under the objective, a region of interest is determined by probing tiles across a designated search area. A single autofluorescence image is acquired at the bottom of each tile and classified based on whether it intersects the sample with a preset intensity threshold. When a tile that intersects the sample is found, neighboring tiles are searched in such a way as to trace the perimeter of the tissue. Tiles inside the perimeter are marked for imaging without being explicitly probed for efficiency. Finally the region of interest is dilated by one tile to ensure that tiles only partially intersecting the tissue sample are also acquired. The search process is finished when all the tiles in the search area have been classified. The marked region of interest is then imaged by sequentially volume-scanning constituent tiles. After all the tiles within the marked region of interest are imaged, the vibratome is engaged and a programmed stage motion feeds the sample through the blade to remove the top layer of tissue. The sample is translated back under the objective such that the newly exposed tissue surface is aligned with the focal plane and the process is repeated until no further tiles are found to image.

Data was acquired in the coronal plane with tile dimensions of 386 × 422 × 250 μm (1024 × 1536 × 251 pixels) and an average dwell time of 61 ns/pixel. eGFP imaging was performed using a tunable ultrafast laser (Chameleon Ultra II, Coherent, Santa Clara, CA) at 920 nm using high excitation intensity (350–400 mW at the objective back aperture).

## Image processing and registration

Image registration was implemented using custom scripts in Matlab (The MathWorks, Natick, MA). For laterally adjacent tiles, no registration was required; translating tiles using recorded stage positions was sufficient. For vertically adjacent tile pairs that spanned a sectioning plane, a set of corresponding features was determined using the descriptor-based image registration plugin in Fiji (*Preibisch et al., 2009*). Because of the large number of tile pairs (>15,000 per brain), this procedure was distributed across a high-performance computing cluster. Compute time was approximately 1 min per tile pair (RAM 128 GB; 16 Intel E5-2680 CPUs). A set of common features, such as punctate auto-fluorescent bodies (lipofuscin; *Figure 3—figure supplement 1*) (*Dowson and Harris, 1981*; *Terman and Brunk, 2004*), restricted to the putative overlap volume was successfully determined in approximately 2/3 of all tile pairs. Many of the remaining tile pairs either included little or no tissue (e.g. around the perimeter of the sample), or they were from relatively featureless areas within the tissue sample (e.g. ventricles and some white matter tracts).

Next, point correspondences were aggregated across each tissue section and translated into the coordinate space defined by the stage system. From these point correspondences, a 3D displacement field was determined by fitting a cubic surface to the point cloud using the RegularizeData3d package for Matlab (http://www.mathworks.com/matlabcentral/fileexchange/46223-regularizedata3d). Although only axially-adjacent tile pairs were registered, tile pairs

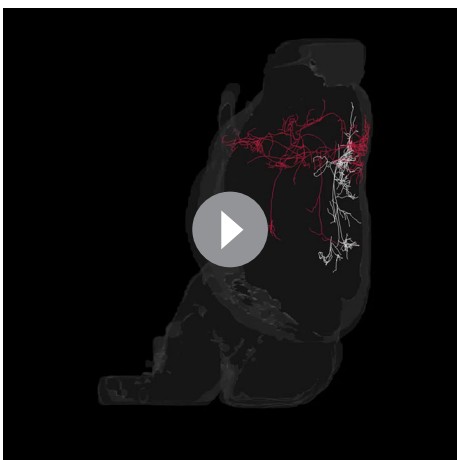

**Video 4.** Three-dimensional rendering of reconstructed Layer V motor cortical neurons. Displayed brain outline corresponds to the contours of the imaged brain. Color code is the same as in all figures.

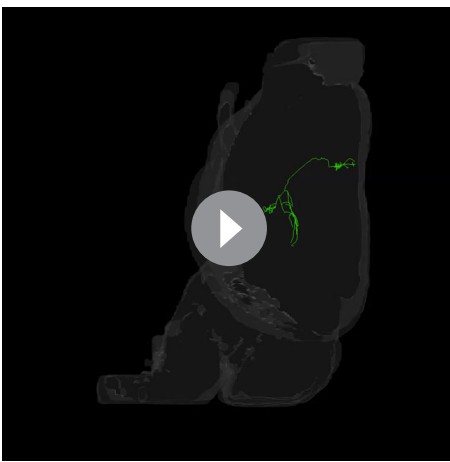

**Video 5.** Three-dimensional rendering of reconstructed Layer VI motor cortical neuron. Displayed brain outline corresponds to the contours of the imaged brain. Color code is the same as in all figures.

could be displaced relative to one another in all three Cartesian coordinates, necessitating the determination of deformation in three dimensions. This 3D displacement field was parameterized laterally for each (x,y) position across the section plane (*Figure 3b*). After the displacement fields between the current section and the sections above and below the current layer were determined, a 3D transform was computed to update the position of each tile within the full imaged volume. To solve for the transform, eight control points were determined for each tile at points offset slightly from the tile corners and located at the centroid of the overlap regions of neighboring tiles. Except in edge cases, there were seven overlapping tiles in the vicinity of each tile corner. These control points formed a regular grid across each section. Sampling the vector field at the centroid of these overlap regions is important for maintaining lateral continuity. We approximated the projection of these points from pixel space into the global coordinate space using a unique affine transform for each tile. Although useful for registration purposes, the presence of lipofuscin was sometimes undesirable during the tracing process even though these puncta were generally straightforward to disambiguate from labeled neurites. Due to the broad emission spectrum of these features, however, they could be computationally removed using standard linear spectral unmixing techniques. After this procedure, unmixed images - - computed from two-channel image tiles - contained only eGFP-labeled structures. Images in *Figures 3–5* and *Video 1* were unmixed according to $U = G - kR$, where $U$ is the unmixed image, $G$ is the green channel of the original image, and $R$ is the red channel of the original image. $k$ is an empirically-determined scale factor.

## Image visualization and annotation

For the purposes of viewing and annotating datasets, all of the tiles in each dataset (input tiles) were resampled into a common coordinate space according to the individual affine transforms determined during the registration procedure. This produced a set of axis-aligned non-overlapping image stacks (output tiles) spanning the imaged volume. Resampling was achieved by back-projecting each voxel in each output tile to one or more input tiles where the nearest voxel was sampled. To avoid aliasing, input tiles were pre-filtered with a Gaussian kernel (sigma = 150 nm). In regions where two or more input tiles overlapped, the maximum intensity was used for the corresponding location in the output tile. The resampling task was implemented on a GPU, which ensured that execution time was dominated by the time required to read input data. Work was distributed output tile-wise over a GPU cluster (20 nodes with 7× GTX 580 1.5GB VRAM). Importantly, each node had parallel access to high-bandwidth network storage. Data were resampled to voxels 0.3 × 0.3 × 1 μm, and stored on disk along with hierarchically downsampled representations of the same volume for visualization at different spatial scales.

In order to smoothly navigate and annotate neuronal structures across large volumes of rendered data, we used a custom plugin for the Janelia Workstation (*Murphy et al., 2014*) that displays and predictively pre-caches multiresolution data using a scheme similar to that employed for large imaging datasets in electron microscopy (*Anderson et al., 2011*; *Cardona et al., 2012*; *Saalfeld et al., 2009*). The neuronal structures depicted in this study were reconstructed by manually placing control points approximately every 5–10 μm and were stored using the SWC file format. Manual reconstruction of neurons required approximately 10–30 min/mm and speed depended upon labeling density, signal intensity, and annotator experience.

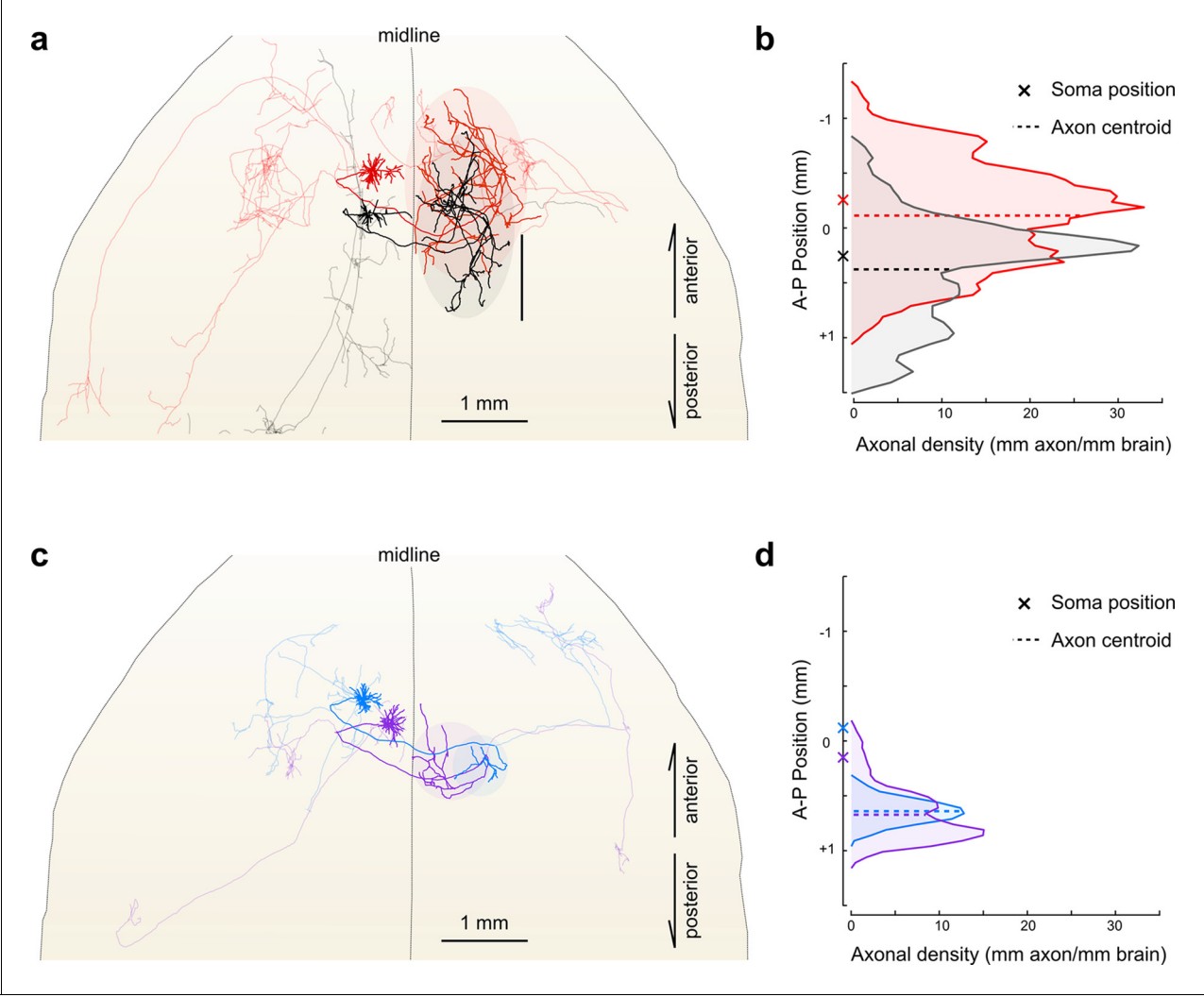

**Figure 7.** Fine scale topology of contralateral cortical projections. (**a**) Horizontal view of the dendrites and axons of two layer V IT neurons (red and black). Axons projecting to locations other than the contralateral motor cortex are shown in lighter colors. (**b**) Density of axonal projections within the contralateral motor cortex as a function of position along the anterior–posterior (A–P) axis. Each point represents the cumulative length of axon within a 200 μm bin centered at the given coordinate divided by the bin width. A–P coordinates are relative to the center of the injection volume. (**c**) Horizontal view of Layer II IT neurons (blue and purple). Same view as in (**a**). (**d**) Density of axonal projections for the two neurons in (**c**) are on the same scale as in (**b**). Color codes are the same as in *Figures 5* and *6*.

## Image compression

To archive datasets, raw data was compressed using the H.264 format (libavcodec; parameters: crf 18, preset slow, tune film). Each color channel was compressed independently as a 16-bit grayscale volume. Compression of a full dataset was distributed, tile-wise, using a high-performance compute cluster. Compressed datasets were typically ~1 TB/channel in size.

## Development of clearing protocol

DMSO has a high refractive index (1.479) and is used extensively in mounting medium formulations for fluorescence microscopy. To improve the clearing properties of DMSO, we investigated solutions of DMSO containing a sugar or sugar alcohol, based on the established tissue-clearing properties of these molecules. We examined the solubility of several compounds: sucrose, D-fructose, D-mannitol, D-sorbitol, xylitol, maltitol, and *meso*-erythritol. We found that D-sorbitol was highly soluble in DMSO up to 1 g D-sorbitol per 1 ml DMSO (44% w/w). Although the saturated solution of D-sorbitol in

**Table 3.** Comparison with alternative technologies for whole-brain imaging.

| Method | Lateral resolution (L; μm) | Axial resolution (A; μm) | L × L × A (μm$^3$) | Speed (× 10$^6$ μm$^3$/s) |
|---|---|---|---|---|
| This study | 0.45 | 1.33 | 0.26 | 1.6 |
| Selective plane illumination microscopy* | 0.65 | 7.30 | 3.10 | 160 |
| Knife-edge scanning microscopy/ micro-optical sectioning tomography (*Li et al., 2010*) | 0.71 | 1.0 | 0.50 | 1.0 |
| Transmission electron microscopy with camera array (*Bock et al., 2011*) | 0.004 | 0.045 | $7.2 \times 10^{-7}$ | $5.6 \times 10^{-6}$ |
| Serial block-face electron microscopy (*Helmstaedter et al., 2013*) | 0.0165 | >0.025 | $>6.8 \times 10^{-6}$ | $1.1 \times 10^{-6}$ |

*Ideal resolution in fully cleared whole mouse brain (P. Keller, personal communication).
*Comparison of resolution and imaging speed with other imaging modalities.* Resolution values represent full width at half maximum except for in transmission electron microscopy with camera array and serial block-face electron microscopy, where pixel sizes are reported.

DMSO was an effective tissue-clearing agent, this organic environment irreversibly decreased the fluorescence of eGFP in tissue slices. We therefore measured the fluorescence of purified eGFP as a function of DMSO concentrations (in 10 mM HEPES, pH 7.3) with or without D-sorbitol (*Figure 2b*). Surprisingly, eGFP fluorescence increased modestly at low concentrations of DMSO followed by a severe decrease at 70% v/v DMSO without D-sorbitol and at 80% v/v DMSO in solutions that included D-sorbitol. We then screened ternary mixtures of DMSO/H$_2$O/D-sorbitol, to produce the clearing protocol described in *Table 1*, which uses increasing amounts of D-sorbitol with a final concentration of 45% w/w D-sorbitol in a solvent made up of 60% v/v DMSO in 0.01 M phosphate buffer. Increasing the concentration of D-sorbitol to its solubility limit (63% w/w; Solution #6 in *Table 1*) further improved clearing, but evaporation and mechanical agitation initiated precipitation of D-sorbitol and/or gel formation during sectioning and imaging.

## Tissue processing and clearing

Mice were deeply anesthetized with an overdose of isoflurane and transcardially perfused with PBS that included 20 U/ml heparin (H3393; Sigma-Aldrich, St. Louis, MO) followed by 4% paraformaldehyde in PBS. Brains were extracted immediately following perfusion and post-fixed for 4 hr at 4°C and washed 3× in 0.1 PBS for >1 hr each. Brains were immersed in CUBIC-1 (*Susaki et al., 2014*) for 3–7 days to remove lipids and subsequently washed 3× in PBS for >1 hr each. Nuclei of samples were optionally counter-stained with 10 μM NuclearID-Red solution (ENZ-52406, Enzo Life Sciences, Farmingdale, NY), washed, embedded in gelatin (12% w/v in PB) and fixed in 4% paraformaldehyde for 12 hr. Cross-linking the tissue to the gelatin embedding medium enabled stable, high SNR imaging even when only a thin section of material remained. Embedded samples were subsequently cleared by immersion in solutions #1–5 (*Table 1*) (1 day in each step). DMSO (472301) and D-sorbitol (85529) were obtained from Sigma-Aldrich. High-purity D-sorbitol was found to be necessary to ensure the absence of fluorescent contaminants. PB and PBS were used at 0.01 M and 0.1 M, respectively.

## Viral labeling

High titer (> 10$^{12}$ GC/ml) AAV 2/1 Syn-iCre and AAV 2/1 CAG-Flex-eGFP were obtained from the Janelia Research Campus Molecular Biology Core. The Cre virus was diluted 1:45,000 in sterile water and combined at a 1:1 ratio with the eGFP virus for injections into the motor cortex of C57/BL6 mice (M/L +0.7; A/P +1.3; D/V −0.75, −0.5, −0.25 relative to bregma; 30 nl/depth). All experimental protocols were conducted according to the National Institutes of Health guidelines for animal research and were approved by the Institutional Animal Care and Use Committee at Howard Hughes Medical Institute Janelia Research Campus (Protocol #14–115).

## Acknowledgements

We thank the Janelia Scientific Computing team, in particular Christopher Bruns and Don Olbris, for help in the development of data visualization tools; the Janelia Instrument Design and Fabrication team, in particular Daniel Flickinger for optical design and construction, Brian Coop for mechanical design, and Magnus Karlsson for custom electronics; Zengcai Guo helped develop the labeling strategy; Monique Copeland and the Janelia Histology Facility provided histological assistance; Jared Rouchard and the Janelia Vivarium staff for surgical assistance and animal care. We thank Nelson Spruston and Adam Hantman for useful discussions and feedback on the manuscript. Scott Sternson, Josh Dudman, Saul Kravitz, Vijay Samalan and Tim Harris provided helpful suggestions. This work was supported by the Howard Hughes Medical Institute and the Janelia MouseLight Project.

## Additional information

### Funding

| Funder | Author |
| --- | --- |
| Howard Hughes Medical Institute | Michael N Economo<br>Nathan G Clack<br>Luke D Lavis<br>Karel Svoboda<br>Jayaram Chandrashekar |

The funders had no role in study design, data collection and interpretation, or the decision to submit the work for publication.

### Author contributions

MNE, LDL, JC, Conception and design, Acquisition of data, Analysis and interpretation of data, Drafting or revising the article; NGC, Conception and design, Analysis and interpretation of data, Drafting or revising the article; CRG, Analysis and interpretation of data, Drafting or revising the article; KS, EWM, Conception and design, Drafting or revising the article

### Ethics

Animal experimentation: All experimental protocols were conducted according to National Institutes of Health guidelines for animal research and were approved by the Institutional Animal Care and Use Committee at Howard Hughes Medical Institute Janelia Research Campus (Protocol #14-115).

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
