## [Decision Letter]

Thank you for submitting your work entitled "A platform for brain-wide imaging and reconstruction of individual neurons" for peer review at *eLife*. Your submission has been favorably evaluated by Eve Marder (Senior editor), a Reviewing editor, and two reviewers.

The reviewers have discussed the reviews with one another and the Reviewing editor has drafted this decision to help you prepare a revised submission.

Summary:

This manuscript describes methods for brain-wide imaging and reconstruction of individual neurons. The authors developed a high-speed two-photon microscope integrated with a tissue vibratome to obtain high-resolution images of whole mouse brains for manual tracing of sparsely labeled neurons. The fast-scanning STP system that can collect images from the entire mouse brain at 0.3 x 0.3 x 1.0 μm resolution in about 1 week of imaging time. This work is timely and potentially quite valuable.

Essential revisions:

1) The authors state: "Although multiple clearing methods have been developed for brain tissue, including CUBIC, CLARITY, Sca/e […], and iDisco, none of these approaches satisfied all of these criteria." It's unclear why all these methods fail to satisfy the criteria. The authors should provide data showing that the other clearing methods are not compatible with their approach.

2) The authors claim that their approach allows for imaging up to 16 million voxels per second (>50 mm^3^ per day), a 16-48 X increase in speed over conventional laser-scanning microscopy systems. A table comparing the throughput of the presented method with those of other high-speed imaging techniques (e.g., SPIM) would be helpful.

3) The authors use their method with GFP. Is their method compatible for brain-wide imaging and tracing neurons that express other fluorescent proteins (mCherry, tdTomato, BFP, etc.)? Can multiple fluorescent proteins (e.g. GFP and mCherry) be imaged and traced within the same brain? These points have to be clarified and backed up with evidence or data.

4) It is important to include images showing lipofusin autofluorescence, perhaps in a supplemental figure.

5) The authors state in the Methods that "for laterally adjacent tiles, no registration was required," but Figure 3 shows conflicting results.

6) What is the typical signal to noise ration of the acquired image? Dwell time is only 61 ns and this corresponds to only 5 excitation pulses per dwell time for 80MHz rep. rate. Does this mean that the system requires highly dense labeling? General limitations of the imaging system need to be discussed.

7) The authors demonstrate bright labeling of axonal termini in Figure 5, but these come from local collaterals nearby the somas. It would be more informative to show the termini of long-range projecting branches to demonstrate the point of high SNR labeling throughout the neuron morphology.

---

## [Author Response]

Essential revisions:

*1) The authors state: "Although multiple clearing methods have been developed for brain tissue, including CUBIC, CLARITY, Sca/e […], and iDisco, none of these approaches satisfied all of these criteria." It's unclear why all these methods fail to satisfy the criteria. The authors should provide data showing that the other clearing methods are not compatible with their approach.*The clearing methods listed were not adopted because one or more properties rendered them fundamentally incompatible with long-term sectioning and imaging. The table below summarizes the specific incompatibilities of each clearing technique with long-term fluorescence imaging integrated with serial sectioning. We have added additional text to the Results section (“Sample clearing and embedding”, first paragraph) more specifically detailing the incompatibilities of each method with our approach.

Nevertheless, these alternative clearing techniques remain useful tools for other applications and our intent is not to highlight their deficiencies. Instead, we only attempt to describe the rationale for the methodological choices made in developing our approach. For this reason, we believe that the revised discussion of alternative clearing techniques in the text is a more appropriate way to strengthen the revised manuscript than the inclusion of this table or additional data. A complete, quantitative evaluation of each clearing technique across all relevant parameters would be a considerable undertaking and it is not clear that this would strengthen the major conclusions of our study. Furthermore, many of the deficiencies that we describe have been confirmed in an additional study published while our manuscript has been under revision (Hama et al., 2015).

**Table d36e5622:** 

	**Cubic**	**Sca*l*e**	**Clarity**	**SeeDB**	**3Disco**	**iDisco**
Quenches fluorescent proteins					x	x
Attenuates fluorescent proteins	x	x	x			
Tissue becomes hard/brittle					x	x
Immersion medium not robust to evaporation	x		x	x		
Requires high viscosity immersion medium	x		x	x		
Introduces high background fluorescence				x		
Not robust to evaporative water loss	x		x	x		
Causes tissue browning over time				x		
Causes issue expansion		x				
Poor white matter clearing		x				
Causes unstable tissue geometry		x				
Tissue soft and highly deformable			x			
Difficult to achieve reproducibility			x			

2) The authors claim that their approach allows for imaging up to 16 million voxels per second (>50 mm^3^ per day), a 16-48 X increase in speed over conventional laser-scanning microscopy systems. A table comparing the throughput of the presented method with those of other high-speed imaging techniques (e.g., SPIM) would be helpful.

The reviewers are correct that a number of imaging techniques may be successfully applied to axonal imaging in large tissue volumes. While these were briefly described and contrasted in the original manuscript, we agree that this comparison was largely qualitative. We have now provided a more detailed, quantitative comparison and have included it as Table 3 in the revised manuscript to augment the treatment of this topic in the text (Discussion, paragraph three). We note that no other technique has been shown to have the required speed, reliability, resolution, and SNR necessary to extensively reconstruct axonal collaterals across the full mouse brain.

3) The authors use their method with GFP. Is their method compatible for brain-wide imaging and tracing neurons that express other fluorescent proteins (mCherry, tdTomato, BFP, etc.)? Can multiple fluorescent proteins (e.g. GFP and mCherry) be imaged and traced within the same brain? These points have to be clarified and backed up with evidence or data.

The reviewers are correct that data in the original manuscript was limited to samples in which eGFP was used as a fluorescent label. In the revised manuscript, we have included an example (Figure 2) demonstrating high-signal-to-noise imaging of intermingled axons labeled with spectrally separable and phylogenetically distinct proteins (eGFP and tdTomato). We also include data demonstrating that tdTomato and other red-shifted DsRed-derived fluorescent proteins (mCherry and mPlum) are not attenuated by our clearing procedure (Figure 2). The ability to trace tdTomato-labeled neurons using our platform does not differ qualitatively from the examples presented with eGFP-labeled neurons. However, complete datasets including multiple spectral labels with additional reconstructions are not central to the description of our platform as a technical resource and will be presented in a subsequent study addressing specific biological hypotheses.

Comparatively dim labels (e.g. BFP) with small two photon cross sections and/or low quantum efficiencies are likely to produce images with reduced signal-to-noise and may not be suitable for imaging fine axonal processes using our platform (or, in fact, most other fluorescence imaging modalities relying on native unamplified fluorescence). Additionally, imaging multiple fluorophores with minimally-overlapping excitation spectra (e.g. BFP or mCherry in addition to eGFP) would require a second excitation source with appropriate spectral characteristics.

While our platform requires bright labels expressed at high levels, eGFP and tdTomato represent two of the most commonly used fluorescent labels and expression using AAV vectors has become a standard technique in neurobiology. We have noted the requirement for bright labeling in the revised Results (“High-resolution whole-brain imaging”, paragraph three).

4) It is important to include images showing lipofusin autofluorescence, perhaps in a supplemental figure.

As requested, we now include an example image in which lipofuscin has not been computationally removed in Figure 3—figure supplement 1.

5) The authors state in the Methods that "for laterally adjacent tiles, no registration was required," but Figure 3 shows conflicting results.

Features lying within the overlap region between axially-adjacent tiles may be displaced relative to their expected position in all three dimensions (X, Y, Z). This three-dimensional displacement between axially-adjacent tiles is illustrated across the two-dimensional (X, Y) exposed surface of the specimen in Figure 3 and is thus consistent with the text. This point is admittedly confusing and has been clarified in the Materials and methods (“Image processing and registration”, paragraph two).

6) What is the typical signal to noise ration of the acquired image? Dwell time is only 61 ns and this corresponds to only 5 excitation pulses per dwell time for 80MHz rep. rate. Does this mean that the system requires highly dense labeling? General limitations of the imaging system need to be discussed.

The reviewers are correct that high signal-to-noise imaging with short dwell times can be very challenging. In the updated manuscript, we have quantified the SNR across randomly selected axonal cross-sections (following Ragan et al., 2012) both before and after image compression (Figure 5—figure supplement 1). Statistics describing the distribution of signal to noise values are now reported in the updated Results (“Reconstruction of long-distance axonal morphology”,paragraph one), but varied widely (peak z-score: 15.0 ± 2.9; median ± s.e.m.; 5-95% range: 7.0–70.7). This variability follows from the heterogeneity of neuronal labeling and, more importantly, the precise spatial location chosen along each axon, as axon diameters vary considerably, even along local (~10 µm) segments. The authors highlight Video 2 as compelling evidence that axons can be imaged with high signal-to-noise along their entire length. The demonstration that high-resolution, high-SNR two photon imaging can be performed with comparatively high speeds is an important novel aspect of our study.

7) The authors demonstrate bright labeling of axonal termini in Figure 5, but these come from local collaterals nearby the somas. It would be more informative to show the termini of long-range projecting branches to demonstrate the point of high SNR labeling throughout the neuron morphology.

Figure 5 illustrates axonal segments (including branch points and termini) far removed from respective somata. The brain structure corresponding to each segment is listed below each image. These structures included the corpus callosum, contralateral motor cortex, piriform cortex, striatum, olfactory tubercle, somatosensory cortex, auditory cortex, and thalamus – all long-range targets of the labeled neurons in the motor cortex. This was not indicated clearly in the original manuscript. We have clarified the figure legend for Figure 5 to make this point more directly.

References:

Hama, H. et al. (2015). ScaleS: an optical clearing palette for biological imaging. Nature Neuroscience, 18(10), 1518–1529. http://doi.org/10.1038/nn.4107.